# Quantitative mapping of transcriptome and proteome dynamics during polarization of human iPSC-derived neurons

Feline W Lindhout[1†], Robbelien Kooistra[1†], Sybren Portegies[1†], Lotte J Herstel[1], Riccardo Stucchi[1,2], Basten L Snoek[3], AF Maarten Altelaar[2], Harold D MacGillavry[1], Corette J Wierenga[1], Casper C Hoogenraad[1,4*]

[1]Cell Biology, Neurobiology and Biophysics, Department of Biology, Faculty of Science, Utrecht University, Utrecht, Netherlands; [2]Biomolecular Mass Spectrometry and Proteomics, Bijvoet Center for Biomolecular Research and Utrecht Institute for Pharmaceutical Sciences, Utrecht University, Utrecht, Netherlands; [3]Theoretical Biology and Bioinformatics, Utrecht University, Utrecht, Netherlands; [4]Department of Neuroscience, Genentech, Inc, San Francisco, United States

**Abstract** The differentiation of neuronal stem cells into polarized neurons is a well-coordinated process which has mostly been studied in classical non-human model systems, but to what extent these findings are recapitulated in human neurons remains unclear. To study neuronal polarization in human neurons, we cultured hiPSC-derived neurons, characterized early developmental stages, measured electrophysiological responses, and systematically profiled transcriptomic and proteomic dynamics during these steps. The neuron transcriptome and proteome shows extensive remodeling, with differential expression profiles of ~1100 transcripts and ~2200 proteins during neuronal differentiation and polarization. We also identified a distinct axon developmental stage marked by the relocation of axon initial segment proteins and increased microtubule remodeling from the distal (stage 3a) to the proximal (stage 3b) axon. This developmental transition coincides with action potential maturation. Our comprehensive characterization and quantitative map of transcriptome and proteome dynamics provides a solid framework for studying polarization in human neurons.

*For correspondence:
c.hoogenraad@uu.nl

[†]These authors contributed equally to this work

## Introduction

Neuronal development is a complex multistep process in which neurons undergo dramatic morphological changes, including axon outgrowth, dendritogenesis, and synapse formation. Much of the fundamental knowledge about neuronal development is based on experimental studies in non-human model systems, such as *Drosophila, C. elegans*, mice and rats (*Zhao and Bhattacharyya, 2018*). However, to what extent the knowledge obtained in these animal models can be extrapolated to human neuronal development remains largely unclear (*Zhao and Bhattacharyya, 2018*). Moreover, analysis of human-specific characteristics of dynamic processes in neurons is hindered by the difficulty in obtaining human brain tissue. The generation of human induced pluripotent stem cells (iPSCs) has provided a critical step forward for studying the development and function of human neuronal cells.

In recent years, many labs have used human iPSC-derived neuronal cultures to study fundamental neurobiological questions. This has contributed to our understanding of processes such as neuronal development, spine development and synaptic plasticity in human neurons. For example, human

iPSC-derived model systems have been used to study dynamic changes in gene expression during early neurogenesis, and to study differentiation of neuronal progenitors (*Compagnucci et al., 2015*; *Grassi et al., 2020*). In addition, human synaptic transmission and plasticity have been studied at single cell level in hiPSC-derived neurons, and human-specific protein functions have been shown to regulate excitatory synaptic transmission specifically in human neurons (*Meijer et al., 2019*; *Marro et al., 2019*). These examples illustrate how the use of human iPSC-derived neurons as a model system can lead to novel findings for human neurodevelopment.

One of the classic model systems to study neuronal development and polarity are the dissociated rat hippocampal neuron cultures developed by Banker and collaborators (*Dotti et al., 1988*). These neurons undergo five well-defined developmental stages, transforming from round, spherical cells to fully mature, polarized neurons (*Craig and Banker, 1994*). First, the symmetric young neurons form small processes (stage 1) and multiple neurites (stage 2). Next, the cells undergo polarization, where one neurite is specified as the axon (stage 3), while the remaining neurites will further develop into dendrites. The axon rapidly extends and further matures during which is assembles an axon initial segment (AIS) at the proximal axon (*Leterrier, 2018* ). The AIS is required for generating action potentials (APs) and maintaining neuronal polarity. In addition to the classic AIS component Ankyrin-G (AnkG), the microtubule binding protein Trim46 also localizes to the AIS and is critical for axon formation by forming parallel microtubule bundles in the proximal axon (*van Beuningen et al., 2015*; *Gumy et al., 2017*; *Harterink et al., 2018*). As the neuron matures, the developing axons and dendrites undergo significant morphological and molecular changes and form dendritic spines (stage 4–5), which allow for the formation of synaptic contacts and the establishment of functional neuron-to-neuron interactions (*Harris and Kater, 1994*; *Fletcher et al., 1994*; *Grabrucker et al., 2009*). In depth proteomic analysis of dissociated rat neurons in culture have identified a number of specific pathways and unique protein profiles that contribute to various aspects of neurodevelopment processes (*Frese et al., 2017*). Proper characterization and quantitative profiling of transcriptome and proteome dynamics is essential to study the specific neurodevelopment events in human iPSC-derived neuronal cultures, including early developmental changes such as neuronal polarization and axon specification.

In this study, we performed extensive characterization of the early developmental stages of hiPSC-derived neurons by immunocytochemistry, electrophysiology, RNA sequencing, and stable isotope labeling combined with high-resolution liquid chromatography-tandem mass spectrometry (LC-MS/MS). We established transcriptomic and proteomic profiles of the early developmental stages (stage 1–3), comprising 14,551 transcripts and 7512 protein identifications, of which we assessed 1163 and 2218 factors that showed differential expression, respectively. These transcriptomic and proteomic profiles point to the importance of microtubule cytoskeleton remodeling in the early stage of neuronal development. Combining this framework with additional methods such as genetic manipulation and live-cell imaging allowed us to investigate the cellular and molecular processes during neuronal polarization and axon outgrowth. Specifically, we identified a distinct, previously unrecognized developmental stage during early axon development, characterized by the reorganization of the axonal microtubule network and relocation of AIS proteins from the distal to proximal axon. The transition through these early axon developmental stages coincided with the time window in which maturation of action potentials occurred. Together, our study provides a quantitative description of transcriptomic and proteomic profiles of hiPSC-derived neuron cultures, which is a rich resource for further analyses of critical signaling pathways during early human neurodevelopment.

## Results

### Characterization of developmental stages in human iPSC-derived neurons

We first systematically assessed if the human iPSC-derived neurons proceed through the initial neurodevelopmental stages, which have previously been described in dissociated rat neurons (*Dotti et al., 1988*). The hiPSC-derived neuron cultures were obtained by neuronal induction of neuronal stem cells (NSCs) and maintained up to ~15 days. Neurons transited through early neurodevelopmental stages 1–3, which was mapped by transducing neurons with FUGW-GFP lentivirus to

visualize cell morphologies, and immunostaining them at different timepoints (day 1, 5 and 14) for proliferation marker Ki67 and NSC marker Nestin to identify NSCs (stage 1), neuron specific markers ß3-Tubulin and MAP2 to identify differentiated neurons (stage 2 and on), and AIS markers AnkG and Trim46 to discriminate between non-polarized (stage 2) and polarized neurons (stage 3) (*Figure 1A, B*). Differentiated neurons adopted a cortical identity, as shown by the appearance of cortical marker Ctip2 in ß3-Tubulin positive cells specifically (*Figure 1—figure supplement 1A*). The transition from NSCs to polarized cortical neurons was confirmed by western blot analyses of the cultures at different timepoints (day 1, 5 and 14): NSC markers Nestin and Sox2 reduced over time, while neuron marker ß3-Tubulin and axon markers Trim46 and Gap43 increased as cells differentiated and polarized (*Figure 1—figure supplement 1B*). Together, these markers indicated a clear developmental transition of human early neurodevelopment over time: stage 1 NSCs (~day 1) differentiated into stage 2 neurons with a characteristic neuronal morphology (~day 5), and subsequently developed axons containing AIS structures (~day 14) (*Figure 1C–E*). Accordingly, the timing of AIS assembly at day ~13–14 coincided with an observed developmental decrease of axon width, whereas dendrite width remained unaffected (*Figure 1—figure supplement 1C,D*). Successful AIS assembly at relatively similar timepoints was confirmed using different neuronal differentiation protocols and NSCs from different donors (*Figure 1—figure supplement 1E,F*). These observations indicate that hiPSC-derived NSCs follow a relatively prolonged neuronal and axonal development compared to dissociated rat neurons, consistent with the protracted development of the human brain (*Dotti et al., 1988*; *Petanjek et al., 2011*). This further supports the emerging evidence revealing species-dependent differences in developmental timing of human and non-human neurons *in vivo* and *in vitro* (*Shi et al., 2012*; *Nicholas et al., 2013*; *Otani et al., 2016*; *Linaro et al., 2019*). We further analyzed the structural organization of the AIS in axons of hiPSC-derived neurons by quantifying the average fluorescence intensity profiles of Trim46 and AnkG (*Figure 1F,G*). Consistent with previous reports in dissociated rat neurons, we found that Trim46 and AnkG localization largely overlapped, with the peak of AnkG intensity located ~6 µm more distally than the peak Trim46 intensity (*Figure 1F,G*; *van Beuningen et al., 2015*). The AIS structure was also enriched for voltage-gated sodium channels (NaV), which strongly overlapped with AnkG structures (*Figure 1—figure supplement 1G*). In summary, these data demonstrate that human iPSC-derived neurons follow the characteristic sequence of developmental stages during neuronal polarization, which occurs at a relatively slower rate than in non-human neurons.

## Action potential firing of polarized human iPSC-derived neurons

To determine whether the polarized human iPSC-derived neurons fire APs, we performed whole-cell patch clamp recordings of stage 3 neurons. We observed AP firing upon positive somatic current stimulation in nearly all recorded neurons (59/61). Succesful AP firing was confirmed in neurons differentiated using other protocols and derived from other donors (*Figure 1—figure supplement 1H, I*). Neuronal excitability was further analyzed in neurons generated using our standard protocol and cell line, by quantifying the number of APs in response to increasing current stimuli (steps of 5 pA; 400 ms) in 54 neurons (*Figure 1H,I*). Of these, 22 neurons (41%) fired multiple times upon higher current stimulation, while 32 neurons (59%) fired only once, independent of stimulus strength (*Figure 1—figure supplement 2A*). Neurons that fired a single AP showed more immature intrinsic cell properties, including depolarized resting membrane potential, lower input resistance, smaller maximum sodium current and smaller after-hyperpolarization (*Figure 1—figure supplement 2B–E*). Spontaneous AP firing or incoming spontaneous synaptic responses were only observed in a few neurons (4/22). This is in line with previous studies reporting that synapse formation typically starts around two weeks after neuronal induction in human iPSC-derived neurons (*Zhang et al., 2013*). Together, the immunofluorescence and electrophysiology data indicate that human iPSC-derived neurons develop functional axons after neuronal polarization.

## Transcriptomic profiling of developing human iPSC-derived neurons

To assess global changes in gene expression during differentiation and neuronal polarization, we next performed an unbiased, in-depth analysis of the transcript expression profiles during early neuronal development. We collected human iPSC-derived neurons at the previously described developmental stages and monitored mRNA expression changes using quantitative population-based

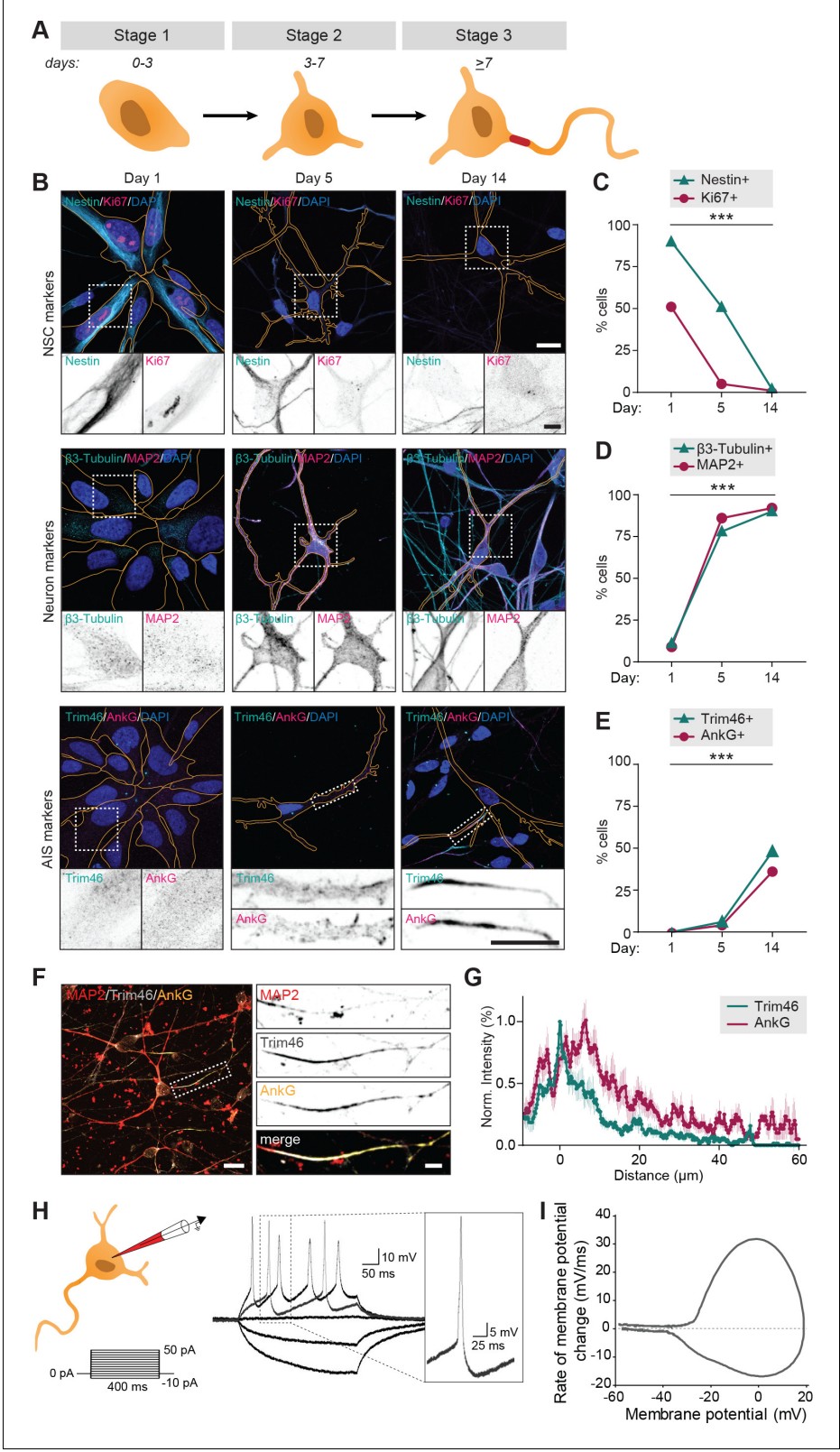

**Figure 1.** Successful and protracted transition through early developmental stages in human iPSC-derived neurons. (**A**) Schematic illustration and timing of neurodevelopmental stages 1, 2 and 3 in human iPSC-derived NSCs/neurons. (**B**) Representative images of stage 1 (day 1), 2 (day 5) and 3 (day 14) hiPSC-derived NSCs/neurons. Cells were transduced with FUGW-GFP lentivirus and immunostained for NSC marker Nestin and proliferation

*Figure 1 continued on next page*

*Figure 1 continued*

marker Ki67, neuron markers β3-Tubulin and MAP2, or AIS markers AnkG and Trim46. Outlines of cells were defined by the FUGW-GFP signal. Scale bar: 15 μm in overview, 5 μm in zooms. (C,D,E) Quantifications of percentage of human iPSC-derived NSCs positive for Ki67 or Nestin (C), β3-Tubulin or MAP2 (D) and AnkG or Trim46 (E) at 1, 5 or 14 days in culture (N = 2, n = 100–109 cells). (F) Representative image of a polarized human iPSC-derived neuron immunostained for MAP2, Trim46 and AnkG. Zoom represents the AIS structure. Scale bar: 20 μm in overview, 5 μm in zoom. (G) Quantifications of average normalized fluorescent intensity profiles for Trim46 and AnkG at proximal axons of human iPSC-derived neurons (day 15) (n = 9). Distances are normalized to Trim46 peak intensities. (H) *Left:* Schematic illustration of the experimental electrophysiology setup. To determine AP frequency, somatic current injections ranging from −10 pA to 50 pA (steps of 5 pA, 400 ms) were applied. *Right:* Representative example of evoked AP firing in a human iPSC-derived neuron, response to hyperpolarizing and two depolarizing current steps, recorded at day 14. *Insert:* first AP to minimal (rheobase) current injection. (I) Phase plot of a single AP of a human iPSC-derived neuron (day 14) that fires multiple APs. NSC: neuronal stem cell, AIS: axon initial segment, AP: action potential. Used tests: Chi-square test (day one vs. day 14) (C–E); ***p<0.001; graphs represent mean ± SEM.

The online version of this article includes the following figure supplement(s) for figure 1:

**Figure supplement 1.** Successful and protracted transition of developmental stages in human iPSC-derived neurons.

**Figure supplement 2.** Characterizing AP firing of early neurodevelopmental stages in human iPSC-derived neurons.

transcriptome analysis. The synchronized differentiation and relatively slow development of these cultures enabled us to select time points at which particular stages manifested in the majority of the cells. The cells were sampled for RNA analysis at days 1, 3, and 7, corresponding to stage 1, the onset of stage 2, and the onset of stage 3, respectively. On the same days, we collected samples for in-depth proteome analysis, which is discussed below. In two biological replicates with two technical replicates each, we identified transcripts corresponding to 14,551 genes by RNA sequencing (*Figure 2—figure supplement 1A*, *Figure 2—source data 1*). Of these, 9655 transcripts were successfully quantified at all time points and normalized to reads per million for further analysis (*Figure 2—source data 2*). As expected, most genes with significantly altered expression were found between day 1 and day 7: 614 genes were downregulated and 549 genes were upregulated at day 7 compared to day 1 (FDR < 0.05 [*Benjamini and Hochberg, 1995*; *Figure 2A*, *Figure 2—figure supplement 1B*, *Figure 2—source data 3*]). Gene ontology (GO) enrichment analysis of downregulated genes indicates that many of the top enriched GO terms relate to processes involved in cell proliferation, such as DNA replication, cell cycle, and cell division (*Figure 2B*, *Figure 2—source data 4*). Upregulated genes correspond to several cellular components and biological processes including nervous system development, neuron projection, and axonogenesis (*Figure 2C*, *Figure 2—source data 4*). Consistently, upregulation of many genes related to neurodevelopment and axonogenesis was previously observed in differentiating mouse embryonic stem cells (ESCs) and iPSCs as well as human ESCs (*Wu et al., 2010*; *Chen et al., 2013*). Interestingly, upregulated genes also showed enrichment of multiple GO terms related to the microtubule cytoskeleton (*Figure 2C*). Among the highly downregulated genes were several proliferation factors, such as SOX2, NOTCH1, and OTX2 (*Figure 2—source data 3*). Furthermore, the chromokinesin motor proteins KIF4a and KIF22, involved in cell proliferation through regulation of spindle microtubule dynamics during mitosis, were downregulated (*Almeida and Maiato, 2018*; *Bisht et al., 2019*). Conversely, genes of several neuronal and axonal kinesins were upregulated, including KIF1a, KIF5c, and KIF21b (*Figure 2—source data 3*; *Hirokawa and Tanaka, 2015*). Additionally, the neuron-specific tubulin TUBB3 and the axonal microtubule associated protein MAPT were upregulated (*Figure 2—source data 3*). The dynamic shift in the transcriptome reflects a population-wide transformation from proliferating cells to terminal differentiating cells with intrinsic neuronal properties, and highlights changes in the microtubule cytoskeleton expression profiles at the onset of stage 3.

## Proteomic profiling of developing human iPSC-derived neurons

In addition to transcriptome analysis, we assessed gene expression on the translational level by performing mass spectrometry based quantitative proteomics. We collected samples for proteome

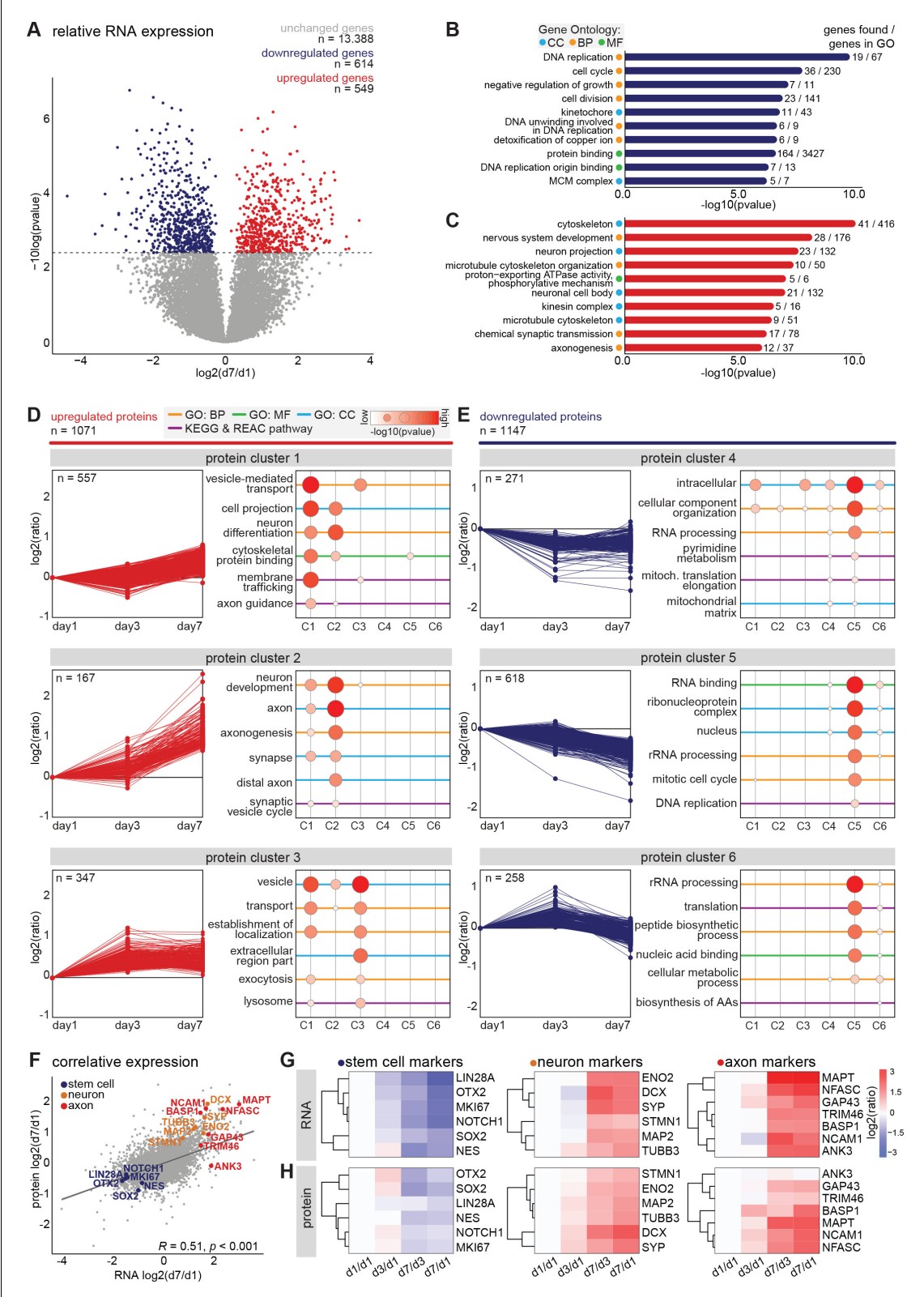

**Figure 2.** Transcriptomic and proteomic profiling of early developmental stages in human iPSC-derived neurons. (**A**) Volcanoplot of differentially expressed transcripts between day 7 and day 1 (false discovery rate (FDR) p<0.05, Benjamini and Hochberg corrected). (**B,C**) Top 10 most significantly enriched GO terms of downregulated (**B**) and upregulated (**C**) genes at day 7. FDR p<0.05, Benjamini and Hochberg corrected. (**D,E**) Six clusters with distinct protein expression profiles, divided in upregulated (**D**) and downregulated (**E**) protein expression, obtained by unsupervised clustering, and the

*Figure 2 continued on next page*

*Figure 2 continued*

GO enrichment analysis for each cluster. (**F**) Correlative analysis of relative transcriptomic and proteomic expression levels (day7/day1) (Pearson's correlation, *R* = 0.51, p<0.0001). Highlighted are selected typical stem cell (blue), neuron (yellow), and axon (red) markers. (**G,H**) Heatmaps showing the relative expression of RNA (**G**) and protein (**H**) levels of typical stem cell, neuron, and axon markers at different timepoints. CC (Cyan): cellular components, BP (Yellow): biological processes, (MF) (Green): molecular function.

The online version of this article includes the following source data, source code and figure supplement(s) for figure 2:

**Source code 1.** source code file script 1 Code bioinformatic analyses.
**Source data 1.** Transcriptomic read counts.
**Source data 2.** Normalized RNA seq read counts (per million reads).
**Source data 3.** RNA seq relative reads per timepoint.
**Source data 4.** RNA seq enriched gene ontologies.
**Source data 5.** Proteomic relative expression and gene ontology enrichment.
**Source data 6.** Comparison RNA and protein expression day 1 versus day 7.
**Figure supplement 1.** Transcriptomic and proteomic profiles of early neurodevelopmental stages in human iPSC-derived neurons.

analysis on the same days as described above for RNA sequencing. We identified 7512 proteins in two technical replicates and quantified 5620 proteins across all three time points (*Figure 2—figure supplement 1C*, *Figure 2—source data 5*). Of these, 2218 proteins showed a changed expression profile during differentiation and neurodevelopment. We assessed the global proteome changes by unsupervised clustering, and identified six clusters with distinct expression profiles (*Figure 2D,E*, *Figure 2—source data 5*). Proteins in clusters 1, 2, and 3 were upregulated, and proteins present in cluster 4, 5, and 6 were downregulated during early neuronal development. Moreover, many proteins within specific clusters showed overlap in functions (*Figure 2D,E*, *Figure 2—source data 5*). Cluster 1 contains proteins that show a slight increase from day 1 to day 7. GO enrichment analysis revealed enrichment of several terms related to neuronal differentiation and intracellular transport mechanisms, which reflects cell-autonomous remodeling of molecular processes (*Figure 2D*, *Figure 2—source data 5*). One of the upregulated proteins in cluster one is KLC1, a subunit of the microtubule motor protein, which was found to be required for neuronal differentiation from human embryonic stem cells (*Killian et al., 2012*). The AIS protein Trim46, which is known to regulate neuronal polarity and axon specification by controlling microtubule organization during development, is also found in this cluster (*van Beuningen et al., 2015*). Furthermore, this cluster contains Camsap1, Camsap2 and Camsap3, proteins which localize to the minus ends of microtubules to stabilize them, thereby regulating neuronal polarity (*Jiang et al., 2014*). Proteins in cluster 2 present a considerable increase in relative expression from day 3 to day 7, which coincides with the onset of axon formation and development (stage 3). Accordingly, enriched GO terms include proteins associated with neuronal development, axonogenesis, and other axon-related mechanisms (*Figure 2D*). Similarly, the GO terms representing factors associated with neuronal development, axons, and synapses were also enriched during differentiation of immortalized human neural progenitor cells (*Song et al., 2019*). Among the highly upregulated proteins in this cluster are several members of the Septin family: neuronal-specific Sept3, Sept5, and Sept6 (*Figure 2—source data 5*). Although mechanistic insights remain unclear, emerging evidence implicates Septins as potential factors for establishing neuronal polarity (*Falk et al., 2019*). Septins interact with actin and microtubule networks and could affect neuronal polarity by regulating cytoskeleton dynamics (*Spiliotis, 2018*; *Falk et al., 2019* ). Sept6 specifically is suggested to play a role in axonal filopodia formation as well as in dendritic branching, and its increased expression coincides with axonal outgrowth (*Cho et al., 2011*; *Hu et al., 2012* ). Moreover, examples of proteins with the highest relative expression in this cluster are DCX, Tau, Ncam1, Basp1, Snap91, and Syt1, which are generally considered to be neuronal differentiation and polarization markers (*Figure 2—source data 5*). These data confirm the neuronal identity of the human iPSC-derived cells, and the presence of cellular machinery involved in axon development. Cluster 3 represents proteins with increased expression from day 1 to day 3, and minimal changes in expression from day 3 to day 7. This cluster comprises proteins enriched in GO terms that are associated with cell metabolism and (re)localization of intracellular and extracellular components, which correspond to substantial changes in the cellular proteome (*Figure 2D*). Proteins in this cluster that show differential expression from day 1 to day 3 include Sox4 and Sox11, both members of the SoxC transcription factor family (*Figure 2—source data 5*). These factors are involved in

neurogenesis and their expression induces subsequent expression of neuron-specific genes (*Kavyanifar et al., 2018*). Also represented in this cluster are Arpc2 and Arpc4, subunits of the Arp2/3 complex. The Arp2/3 complex mediates actin polymerization and is required for formation of lamellipodia and filopodia as well as axon guidance (*Chou and Wang, 2016*). Clusters 4, 5, and 6 encompass proteins that are downregulated during the differentiation of NSCs into polarized neurons. GO analysis of these clusters reveals that they contain proteins involved in intracellular metabolism and homeostasis, genomic translation, the cell cycle, and biosynthesis of amino acids and peptides (*Figure 2E*). Downregulation of DNA replication and cell cycle-related proteins is also reported to coincide with terminal differentiation in neuroblastoma cells and with development of cultured rat neurons (*Murillo et al., 2017*; *Frese et al., 2017*). These results suggest that the overall proteome dynamics are indicative of cellular processes such as cell cycle exit and neuronal differentiation. Together, these dynamic proteomic profiles, similar to the transcriptomic data, represent a population-wide shift from hiPSC-derived NSCs to polarized neurons over time.

## Comparison of transcriptomic and proteomic profiles of developing human iPSC-derived neurons

To compare the transcriptome and proteome dynamics, we performed correlative analysis of the relative RNA and protein expressions on day 7 compared to day 1 (*Figure 2—source data 6*). Based on their annotated gene names we were able to compare the expression dynamics of 7021 factors. Of these, 4536 followed the same trend for transcriptomic and proteomic expression dynamics, and overall, we found a significant correlation between the relative transcriptomic and proteomic expression profiles (*Figure 2F*). In agreement with the observed immunofluorescence, typical stem cell markers (NOTCH1, SOX2, MKI67, LIN28A, OTX2, and NES) showed a downregulation of both RNA and protein levels during neuronal differentiation. RNA levels as well as protein levels of typical neuron markers (DCX, ENO2, SYP, MAP2, STMN1, and TUBB3) and of axonal markers (TRIM46, MAPT, BASP1, ANK3, NCAM1, GAP43, and NFASC) displayed a marked increase during neuronal development (*Figure 2G,H*). Dynamic developmental expression profiles were also observed for several semaphorins and Rap1GTPases, which are important regulators of neuronal polarity and migration in the developing mammalian neocortex (*Figure 2—figure supplement 1D,E*; *Pasterkamp, 2012*; *Shah et al., 2017*; *Wang et al., 2018*). Through quantitative analysis of transcriptomic and proteomic dynamics we were able to characterize human iPSC-derived neuronal differentiation and identify early neurodevelopmental processes in an unbiased manner. These quantitative maps of neuronal transcriptome and proteome dynamics provide a rich resource for further analyses and may identify molecular mechanisms involved in neuronal polarity and axon specification.

## Identification and characterization of intermediate stages during axon specification

Transcriptomic and proteomic profiling of developing neurons revealed that axonal components are upregulated after ~7 days, and assembled AIS structures were detected at proximal axons after ~14 days. Next, we studied the process of axon formation in human iPSC-derived neurons in more detail, and investigated the appearance of AIS proteins at different timepoints between day 5 and 11. In stage 2 neurons, in which neurites are unpolarized and have similar lengths, AIS proteins Trim46 and AnkG appeared as punctate structures at one or more neurites in a subset of neurons (*Figure 3A,B*). Quantification of the relative abundance of stage 2 neurons lacking AIS proteins (referred to as stage 2a), or containing AIS proteins at one or more neurites (referred to as stage 2b), showed a developmental transition over time from stage 2a to stage 2b neurons (*Figure 3A,B*). Stage 3 neurons were morphologically defined by the appearance of a single elongated neurite, the future axon, which has grown at least twice as long as the other neurites. Interestingly, we found that AIS proteins in stage 3 neurons first appeared as noncontinuous structures consisting of multiple smaller puncta and stretches that cover distal parts of the axon (referred to as stage 3a), prior to their more conventional localization in the AIS at the proximal axon (referred to as stage 3b) (*Figure 3A,B*). This newly identified axon developmental stage 3a was also observed in neurons differentiated using other protocols and in neurons derived from other donors (*Figure 3—figure supplement 1A,B*). Quantifications of the abundance of these neurodevelopmental stages over time revealed a developmental decline of stage 2b neurons that was accompanied with an increase of stage 3a neurons, as well as

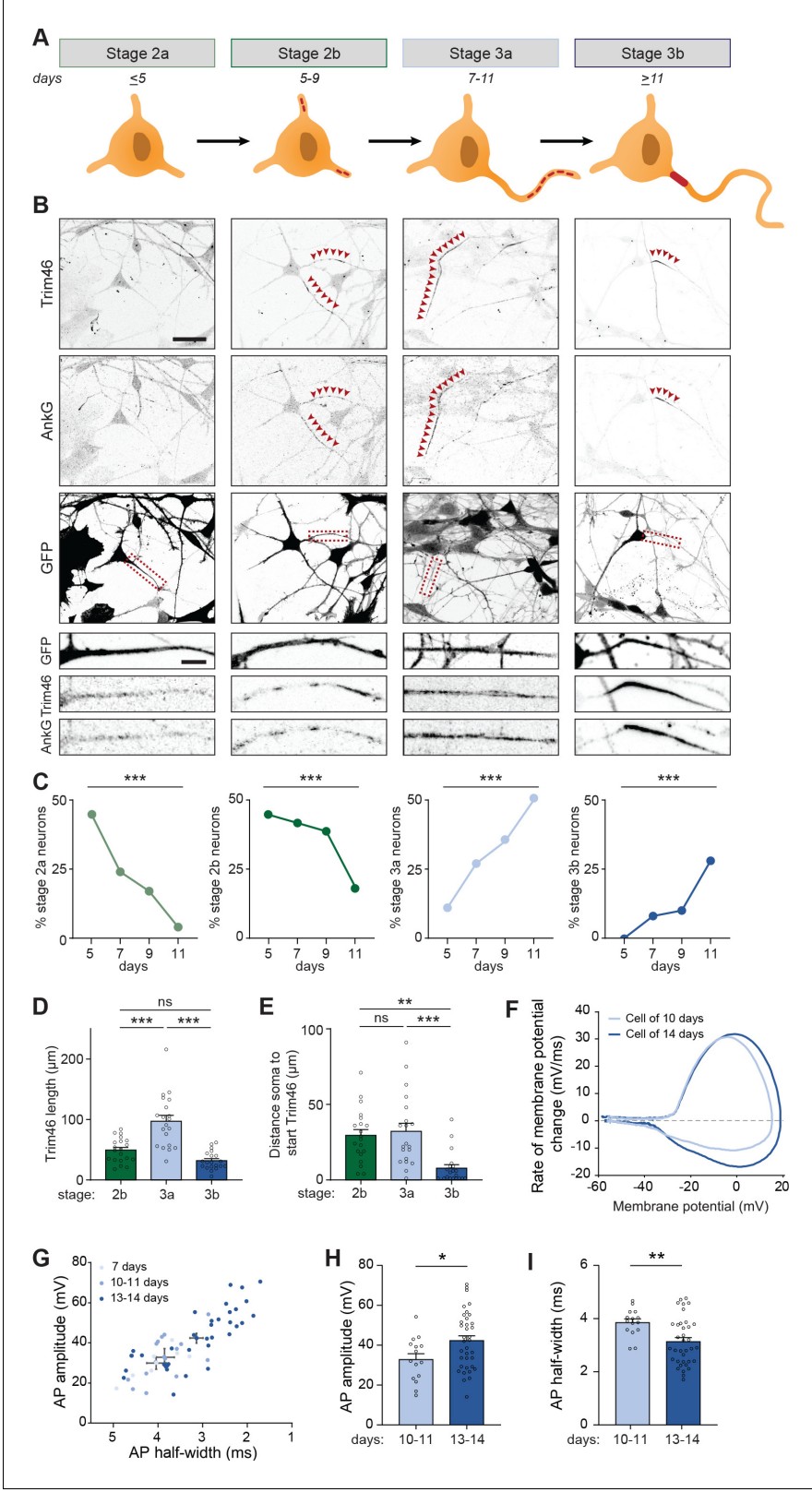

**Figure 3.** Identification of intermediate developmental stages during onset of axon formation. (**A**) Schematic illustration and timing of neurodevelopmental stages 2a, 2b, 3a and 3b in human iPSC-derived NSCs/neurons. (**B**) Representative images of stage 2a, 2b, 3a, and 3b hiPSC-derived neurons. Cells were transduced with FUGW-GFP lentivirus and immunostained for AnkG and Trim46. Arrowheads mark Trim46 and AnkG accumulations. Zooms
*Figure 3 continued on next page*

*Figure 3 continued*

represent a nonpolarized neurite in a stage 2 neuron or a developing axon in a 3 neuron. Scale bar: 40 µm overview, 5 µm zooms. (**C**) Quantifications of the relative abundance of stage 2a, 2b, 3a or 3b human iPSC-derived neurons (N = 2, n = 50–55 cells). (**D**) Quantifications of the total length of Trim46 structures in neurites of stage 2b, 3a and 3b human iPSC-derived neurons (N = 2, n = 20 cells). (**E**) Quantifications of distance from soma to start of the Trim46 signal in neurites of stage 2b, 3a and 3b human iPSC-derived neurons (N = 2, n = 20 cells). (**F**) Phase plots of a representative AP recorded of a human iPSC-derived neuron at 10 days and 14 days. (**G**) Scatter plot of AP amplitude versus AP half-width grouped by days after plating (N = 4; 7 days: n = 7 cells, 10–11 days: n = 15 cells, 13–14 days: n = 36 cells). (**H**) AP amplitude recorded in human iPSC-derived neurons of 10–11 days (N = 4, n = 15 cells) and 13–14 days (N = 4, n = 36 cells). (**I**) AP half-width recorded in human iPSC-derived neurons of 10–11 days (N = 4, n = 15 cells) and 13–14 days (N = 4, n = 36 cells). AP: Action potential. Used tests: Chi-square test with Bonferroni post-hoc correction (**C**), One-way ANOVA with Bonferroni post-hoc correction (**D, E**), Student's t-test (**H**), Mann-Whitney U test (**I**), ***$p < 0.001$, **$p < 0.01$, *$p < 0.05$, ns $p \geq 0.05$; graphs represent mean ± SEM. The online version of this article includes the following figure supplement(s) for figure 3:

**Figure supplement 1.** Extra developmental stage and gradual action potential maturation during axon formation.

an increase of stage 3b neurons with a relatively later onset (*Figure 3C*). We further characterized the distribution of AIS proteins by measuring their lengths and distances to the soma at each stage. Developmental changes in the length of AnkG and Trim46 structures were observed, as the total neurite length covered by Trim46 or AnkG signals was strongly increased by ~40% in stage 3a neurons, and significantly reduced by ~55% in stage 3b neurons (*Figure 3D*; *Figure 3—figure supplement 1C*). Movereover, the axonal Trim46 and AnkG structures were localized more distally in stage 3a neurons, as the distance from the soma to both the start as well as the end of the Trim46 and AnkG appearance was significantly larger compared to stage 3b neurons (*Figure 3E*; *Figure 3—figure supplement 1D–F*). The localization of NaV at axons shows a similar dynamic profile as Trim46 and AnkG during development (*Figure 3—figure supplement 1G*). Together, these data imply that axon specification (transition stage 2–3) in human iPSC-derived neurons can be subdivided in four steps (stage 2a, 2b, 3a and 3b) based on the subcellular localization of AIS proteins. Specifically, AIS proteins first form relatively long, noncontinuous structures in the distal axon (stage 3a) before accumulating at the proximal axon to form the AIS structure (stage 3b).

## Action potential maturation coinciding with onset of axon development

We next investigated whether the different organization of AIS components in stage 3a and 3b neurons is accompanied by differences in their electrical properties. Local clustering of voltage-gated ion channels at AIS structures, as observed in stage 3b neurons, is important to facilitate mature APs (*Kole et al., 2008*). Hence, we hypothesized that the noncontinuous appearance of NaV channels at distal axons in the newly identified stage 3a neurons affect AP firing. To address this, we performed electrophysiological recordings of neurons from day 7 to 14 to capture the developmental transition from stage 3a to stage 3b neurons. Our recordings showed progressive maturation of physiological properties during this developmental time window (*Figure 3F,G*). Neurons recorded on day 13–14 fired APs with a larger amplitude and shorter half-width compared to 10–11 days old neurons (*Figure 3H,I*). No differences in AP amplitude and half-width were found between neurons of 7 and 10–11 days. However, on day 7, neurons fired APs with smaller after-hyperpolarization (data not shown), possibly reflecting a developmental increase in potassium channels (*Song et al., 2013*). Other intrinsic properties, like resting membrane potential, input resistance, AP firing threshold and maximum sodium current, remained stable during this developmental period. Together, these results indicate a developmental maturation of specific AP properties, which coincides with the timing of the developmental transition from stage 3a to stage 3b neurons.

## Mapping microtubule reorganization in the newly identified developmental stages

The transcriptomic and proteomic profiles point to the importance of microtubule cytoskeleton remodeling during axon specification. Indeed, microtubule dynamics and remodeling have been extensively studied, and the axonal microtubule cytoskeleton is important for regulating AIS dynamics (*Stepanova et al., 2003*; *Kleele et al., 2014*; *van Beuningen et al., 2015*; *Yau et al., 2016*).

Therefore, we assessed changes in the microtubule network in axons and dendrites of developing human iPSC-derived neurons by systematically analyzing plus-end dynamics and orientations of microtubules at different locations (*Figure 4A*). We performed two-color live-cell imaging to visualize neuronal morphology and microtubule plus-end tracking proteins (MT+TIPs), respectively (*Figure 4B,C*; *Video 1*). Bidirectional MT+TIP movement, as shown by comets moving in both the anterograde and retrograde direction, was observed at day 5 (representing stage 2 neurons) with a preference for the anterograde direction (*Figure 4D*). Over time, this preference shifted towards more retrograde movement in developing dendrites, and towards unidirectional anterograde movement in developing axons (*Figure 4D*). These changes are consistent with differences in microtubule organization in axons and dendrites found in non-human neurons (*Yau et al., 2016*; *Schätzle et al., 2016*). In similar fashion to the distal to proximal reorganization of AIS proteins (*Figure 3*), the observed shift towards a unidirectional microtubule organization in axons occurred first in distal parts of the axon, followed by proximal reorganization at later time points. The observed developmental changes in anterograde and retrograde ratios are mostly explained by changes in retrograde comets, as the total number of retrograde comets over time was increased in dendrites and decreased in axons (*Figure 4E,F*). The comet growth speed is slightly reduced during development, while comet run length increased markedly, suggesting a reduction in catastrophe events during axon and dendrite development (*Figure 4G,H*; *Figure 4—figure supplement 1A,B*). Imaging of MT+TIPs provides information about the dynamic ends of microtubules, but the fraction of moving comets does not directly reflect the orientations of all microtubules (*Yau et al., 2016*). To analyze microtubule orientations of both dynamic and stable microtubules, we combined our previous live-cell imaging approach with laser-severing. Cutting microtubules with a short-pulsed laser generates new microtubule ends, which allows for analysis of newly formed MT+TIPs (*Figure 4I,J*; *Figure 4—figure supplement 1C,D*; *Video 2*). Following laser severing, a strong shift towards a balanced, bidirectional orientation in developing dendrites is observed (*Figure 4K*). The number of comets in both directions increased during development, with those moving in the retrograde direction increasing relatively more (*Figure 4L,M*). This suggests the presence of a larger pool of stable, minus-end out microtubules in dendrites. In developing axons, the shift towards unidirectional, plus-end out microtubules was more notably observed following laser severing (*Figure 4K*). Laser severing led to a large increase in the number of anterogradely moving comets (*Figure 4L*), suggesting the presence of a stable pool of plus-end out microtubules. These results indicate that the axonal microtubule remodeling in a distal to proximal fashion coincides with the observed relocation of AIS proteins. Together, these data show that axon formation is characterized by the reorganization of the axonal microtubule network and relocation of the AIS from the distal to proximal axon (*Figure 4—figure supplement 1E*).

## Discussion

To better understand neuronal differentiation and polarization in human cells, we performed an in-depth characterization of human iPSC-derived neurons during these developmental processes. We systematically assessed the early stages of human neurodevelopment in culture, including axon specification (transition stage 2–3), and performed transcriptomic and proteomic profiling during these steps. We describe previously unrecognized intermediate stages of axonal outgrowth, which is in particular characterized by a distal (stage 3a) to proximal (stage 3b) reorganization of the axonal microtubule network and relocation of AIS proteins.

### Development of polarized and functional human iPSC-derived neurons

In this study we showed that NSCs consistently gave rise to polarized and functional human neurons, which was demonstrated by the loss of cell proliferation and NSC markers, and the appearance of neuron and AIS markers. These neurons formed an axon with a functional AIS, and exhibited AP firing. As expected, AP properties and cell intrinsic variables appeared immature compared to other studies performed at later developmental stages (*Bardy et al., 2016*; *Gunhanlar et al., 2018*). Passive physiological properties were comparable to neurons recorded from *ex vivo* fetal cortical brain tissue (*Moore et al., 2009*). We consistently observed neurons that fired a single AP and neurons that fired multiple APs at different developmental stages. The underlying difference in these subgroups remain unknown, but may indicate variation in the maturation, cell morphology or other

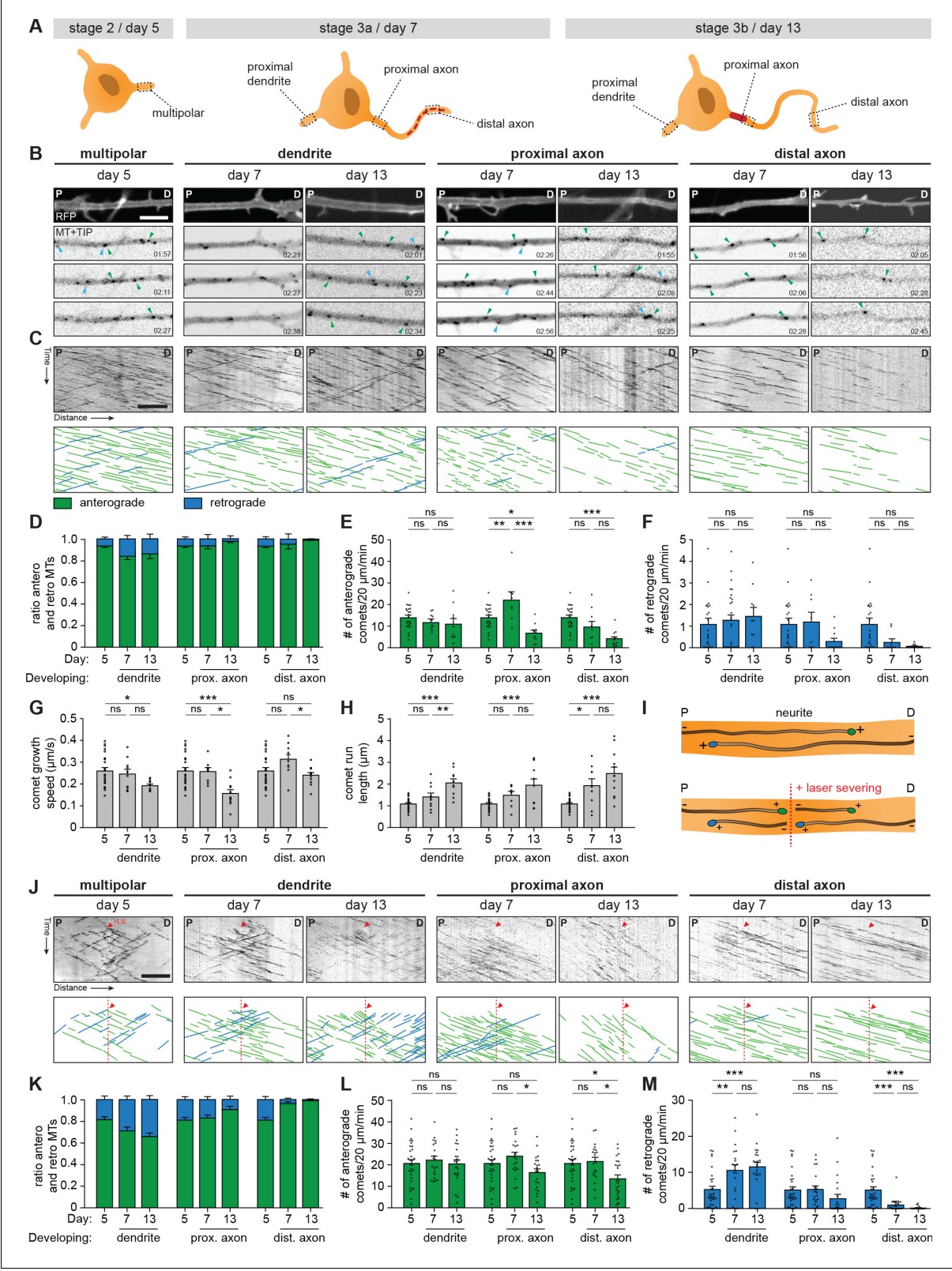

**Figure 4.** Axonal microtubule cytoskeleton is reorganized in a distal-to-proximal fashion during development. (**A**) Schematic illustration of stage 2 (day 5), 3a (day 7) and 3b (day 14) human iPSC-derived neurons. Different locations of the neurons that are characterized for experiments are outlined and annotated. (**B**) Stills from a spinning-disk time-lapse recording of specified neurites transfected with MARCKS-tagRFP_IRES_GFP-MACF18 at specific time points. The top panel is a still of a typical neurite in MARCKS-tagRFP. The other panels show moving GFP-MT+TIP comets (GFP-MACF18) moving

*Figure 4 continued on next page*

*Figure 4 continued*

either anterogradely (green arrowheads) or retrogradely (blue arrowheads). P and D indicate the proximal and distal direction of the neurite, respectively. Scale bar: 5 µm. (C) Kymographs and schematic representations of time-lapse recordings shown in (B). Scale bar: 5 µm. (D) Quantifications of the ratios of comets moving anterogradely (green) or retrogradely (blue) direction (N = 3, n = 8–23 cells). (E, F) Quantifications of the number of comets per minute moving anterogradely (E) and retrogradely (F) (N = 3, n = 8–23 cells). (G, H) Quantifications of the growth speed (G) and run length (H) of comets (N = 3, n = 8–23 cells). (I) Schematic representation of microtubule LS experiments. (J) Kymographs and schematic representations of time-lapse recordings of LS experiments shown in *Figure 4—figure supplement 1b*. Red arrowheads and dotted lines indicate when LS is performed. Scale bar: 5 µm. (K) Quantifications of the ratios of comets moving anterogradely (green) or retrogradely (blue) direction, 10 µm before and after the LS position (N = 3, n = 20–30 cells). (L, M) Quantifications of the number of comets per minute moving anterogradely (L) and retrogradely (M), 10 µm before and after the LS position (N = 3, n = 20–30 cells). LS: laser-severing. Used tests: One-way ANOVA with Tukey's post-hoc analysis (E–H, L, M); ***p<0.001, **p<0.005, *p<0.05, ns p≥0.05; graphs represent mean ± SEM.

The online version of this article includes the following figure supplement(s) for figure 4:

**Figure supplement 1.** Microtubule remodeling in axons and dendrites during early neuronal development.

factors that could contribute to cellular heterogeneity of the culture. Non-human neurons develop at faster rates: for example, rat dissociated neurons reach stage 3 after approximately 1.5 days in culture, and cortical development and maturation in mammals ranging from mouse to primate is both faster and less complex than in humans (*Dotti et al., 1988*; *Clowry et al., 2010*; *Molnár and Clowry, 2012*; *Silbereis et al., 2016*; *Marchetto et al., 2019*). We found that human iPSC-derived neurons transit to stage 3 in approximately 7–14 days, which is consistent with the described prolonged development of human neurons *in vivo* and *in vitro* (*Grabrucker et al., 2009*). The slow rate of development is also reflected by the lack of spontaneous AP firing or incoming synaptic responses as mature synapses have likely not formed yet. Notably, co-culturing with astrocytes could promote synaptogenesis, as this has been shown to enhance synaptic connectivity (*Tang et al., 2013*). The prolonged development of human iPSC-derived neurons allows studying neurobiological processes, such as AP maturation and AIS assembly, with higher temporal resolution.

## Quantitative profiles of transcriptomic and proteomic of early human neurodevelopment

Quantitative transcriptome analysis identified 549 upregulated genes and 614 downregulated genes during early stages of neurodevelopment. As expected, and in agreement with changes observed in other differentiating cell types, GO term enrichment analysis showed strong downregulation of genes related to cell proliferation during terminal differentiation (*An et al., 2014*; *Gao et al., 2014*; *Tripathi et al., 2014*). Simultaneously, genes related to neurodevelopmental processes such as neurite formation and axonogenesis are upregulated, which is in line with changes previously observed in mouse iPSCs and human ESCs (*Wu et al., 2010*; *Chen et al., 2013*). To determine if regulatory changes of transcripts are reflected in protein expression, we performed quantitative proteome analysis. We identified 2218 proteins with more than two-fold expression changes during the first stages of neurodevelopment, indicating that significant remodeling of the proteome takes place during early stages. We identified six clusters of expression profiles and conducted GO enrichment analysis, which revealed a coordinated proteomic rearrangement during neurodevelopment. Proteins related to neuronal differentiation are upregulated and proteins related to cell proliferation are downregulated. Similar changes are observed in dissociated rodent neurons, differentiating neural crest and immortalized human neural progenitor cells, confirming the neuronal identity adopted by our human iPSC-derived cells (*Kobayashi et al., 2009*; *Frese et al., 2017*; *Song et al., 2019*). Similar to changes in protein expression levels, RNA expression also showed a strong increase of factors involved in neuronal differentiation and polarization, such as TUBB3 and TRIM46. Indeed, RNA and protein dynamics were generally correlated, indicating a coordinated cellular reprogramming of NSCs into polarized neurons. Furthermore, reorganization of the microtubule

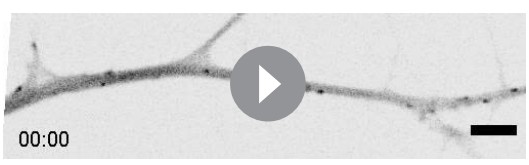

**Video 1.** Representative movie of MT+TIP comet dynamics in a dendrite (day 7). Scale bar: 5 µm. Time in minutes:seconds.
https://elifesciences.org/articles/58124#video1

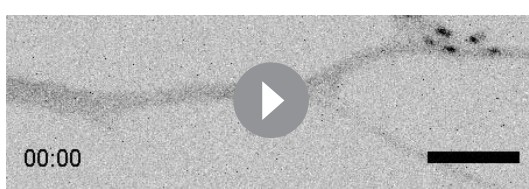

**Video 2.** Representative movie of MT+TIP comet dynamics following laser severing in a dendrite (day 7). Scale bar: 5 µm. Time in minutes:seconds. https://elifesciences.org/articles/58124#video2

cytoskeleton is reflected in both transcriptome and proteome profiles during early neurodevelopment. However, differences between transcriptome and proteome dynamics during early neurodevelopment are also observed. This may reflect a temporal shift in regulation of transcription and translation. In addition, possible regulatory mechanisms on the protein level might be at play. Together, these data provide a rich resource for both the transcriptome and proteome dynamics in developing human neurons, which can be used in future studies to advance our understanding of the molecular mechanisms involved in neuronal differentiation and polarity.

## Distal to proximal relocation of AIS proteins during axon development

A critical event in early neurodevelopment is the polarization of a symmetric cell into a neuron, which is initiated by the formation of a single axon (*Dotti et al., 1988*; *Craig and Banker, 1994*). Here, we studied the onset of polarity in human iPSC-derived neurons in detail, and found that axon specification proceeds through a multistage process with several intermediate steps. We observed that AIS proteins Trim46, AnkG and NaV first appear as long, noncontinuous structures in distal regions of axons at the onset of stage 3 (stage 3a), and later relocate and cluster at proximal regions to form the stable AIS structures (stage 3b). In conjunction, we also found a distal-to-proximal reorganization of the microtubule cytoskeleton network in axons. This is marked by a developmental shift towards the characteristic uniform, plus-end out orientation in growing axons. Although the precise relation between AIS formation and microtubule remodeling in early axon development is unclear, various studies have shown the cooperative interaction between AIS components and microtubule structures during axon development (*Leterrier et al., 2011*; *Fréal et al., 2016*). Recently *Fréal et al., 2019*, described a feedback-based mechanism that drives AIS assembly, in which membrane, scaffolding, and microtubule(-associated) proteins, including AnkG and Trim46, cooperate to form a stable AIS-microtubule structure in the proximal axon. It has been shown that AnkG can act as a scaffold to recruit Trim46-positive microtubules and subsequently direct AIS protein trafficking to the proximal axon (*Fréal et al., 2019*). The possible function of AIS proteins in the distal axon remains elusive. Similar to their function in stabilizing microtubules at the proximal axon, the AIS proteins present at the distal axon may also assist in locally stabilizing microtubules to presumably drive axon outgrowth. AnkG in distal axons may provide additional support to the formation of Trim46-positive parallel microtubules during outgrowth. Then, as Trim46 moves proximally, the axonal microtubule network close to the cell body is also remodeled to the characteristic uniform plus-end out orientation. This idea is consistent with the observed shift of the unidirectional parallel microtubule organization in the distal parts of the axon first, followed by proximal reorganization. Finally, AnkG and Trim46 together may drive AIS assembly at the proximal axon, as previously described in dissociated rodent neurons (*Fréal et al., 2019*). It remains unknown if the observed intermediate step of distal AIS protein accumulation is unique to human neurons. Axons in humans grow significantly longer compared to rodents, thus additional regulatory mechanisms enhancing axon outgrowth might be at play in humans. Alternatively, it is possible that these changes have not been observed in rodent neurons because of their relatively faster development. We also observed differences in microtubule dynamics, as MT+TIP growth speeds are higher than those found in rodent neurons *in vitro* and *in vivo*, hinting at species-specific regulation of microtubule dynamics in human neurons (*Stepanova et al., 2003*; *Kleele et al., 2014*; *Yau et al., 2016*). Future studies are required to examine the potential human-specific attributes of neuronal polarity and axon outgrowth.

In summary, our quantitative map of neuronal transcriptome and proteome dynamics provides a rich resource for future analysis of early neurodevelopmental processes in human iPSC-derived neurons. We investigated early development in human neurons and uncovered an intermediate axon developmental step in stage 3 neurons, thereby illustrating the potential of this model system to study neurobiological processes in human neurons. This study also provides a framework and

important starting point for further studies that aim to complement our understanding of polarization in human neurons.

## Materials and methods

### hiPSC-derived neuronal cell culture

Human iPSC-derived cortical neural stem cells (NSCs; ax0016, Axol Bioscience) were obtained from three different donors (donor #1: ax0016; donor #2 ax0018; donor #3 ax0015, Axol Bioscience). For expansion of hiPSC-derived NSCs (only for ax0016, Axol Bioscience), cells were thawed and quenched with Neural Expansion-XF Medium (ax0030, Axol Bioscience), centrifuged (200 $g$, 5 min), resuspended in Neural Plating-XF medium (ax0033, Axol Bioscience), and plated (~500 k per well) on six-wells plates pre-coated with freshly-thawed SureBond (ON, 37°C; ax0041, Axol Bioscience) in PBS at 37°C with 5% $CO_2$. Medium was replaced on the next day by Neural Expansion-XF Medium supplemented with EGF (20 ng/ml; AF-100–15, Peprotech) and FGF (20 ng/ml; 100-18B, Peprotech). Medium was refreshed every two days, and cells were passaged when cultures reached a 70–80% confluency. For passaging, cells were washed once with PBS, incubated with Unlock (5 min, 37°C; ax0044, Axol Bioscience), quenched with Neural Expansion-XF Medium, and centrifuged (200 $g$, 5 min). Cell pellets were resuspended in Neural Plating-XF medium and plated in a 1:3 ratio on six-wells plates pre-coated with freshly-thawed SureBond (see above). After a maximum of three passaging rounds, cells were frozen in KnockOut Serum Replacement (10828028, Life Technologies) with 10% DMSO and stored in liquid nitrogen. Unless stated differently, neurons were obtained by differentiating hiPSC-derived NPCs from donor #1 (ax0016, Axol Bioscience) following induced synchronous neuronal differentiation (Human iPSC-derived Neural Stem Cells, Protocol version 5.0). Cells were thawed and quenched with Neural Expansion-XF Medium, centrifuged (200 $g$, 5 min), resuspended in Neural Plating-XF medium, and plated on 12 mm (~100 k per well) or 18 mm (~200 k per well) pre-coated glass coverslips in respectively a 24-wells or 12-wells plate at 37°C with 5% $CO_2$. Coating of coverslips was performed directly before plating: coverslips were first incubated with ReadySet (45 min, 37°C; ax0041+, Axol Bioscience), washed four times with sterilized water, and incubated with freshly-thawed 1x SureBond (1 hr, 37°C; ax0041+, Axol Bioscience) in PBS. After 24 hr (day 1), the medium was fully replaced by Neural Maintenance-XF Medium (ax0032, Axol Bioscience), and after another 24 hr (day 2) by Neural Differentiation-XF Medium (ax0034, Axol Bioscience). Three days later (day 5), half of the medium was replaced by Differentiation-XF Medium. Next day (day 6), half of the medium was replaced by Neural Maintenance-XF Medium, again one day later (day 7), and every three days during further maintenance. For the induced synchronous neuronal differentiation protocol #2 (Enriched Cerebral Cortical Neurons User Guide, Axol Bioscience), cells were plated in adjusted Plating Medium composed of Neural Plating-XF medium supplemented with 10 µM Y-27632 (10–2301, Focus Biomolecules). After 24 hr (day 1), and at day 3 and 5, all medium was replaced by adjusted Differentiation Medium composed of Neural Maintenance Basal Medium (ax0031b, Axol Bioscience) supplemented with 1:50 NeurOne Supplement A (ax0674a, Axol Bioscience). At day 7, day 9, day 11 and day 13, all medium was replaced by Maturation Medium composed of Neural Maintenance Basal Medium supplemented with 1:50 NeurOne Supplement B (ax0674b, Axol Bioscience), 20 ng/mL BDNF (248-BDB, R and D Systems), 0.5 mM cAMP (D0260, Merck-Sigma) and 0.2 mM ascorbic acid (A4403, Merck-Sigma). For the spontaneous neuronal differentiation protocol (Spontaneous Neuronal Differentiation User Guide Version 1.0, Axol Bioscience), cells were plated in adjusted Plating Medium and after 24 hr (day 1) all medium was replaced by Neural Maintenance-XF Medium. Every two days during further maintenance, half of the medium was replaced by Neural Maintenance-XF Medium. Cells were kept in culture for maximum ~15 days to ensure high quality of the cultures.

### Lentivirus and lentiviral infection

The constructs used for lentiviral transduction are FUGW-GFP (Addgene #14883; *Lois et al., 2002*) and Marcks-tagRFP-T-pIres-GCN4-MacF18. Marcks-tagRFP-T-pIRES-GCN4-MacF18 cloning is previously described (*Yau et al., 2016*). The construct was subcloned into the pSIN-TRE-mSEAP-hPGK-rtTA2sM2 lentiviral vector (kindly provided by Dr. Didier Trono, Ecole Polytechnique Fédérale de Lausanne, Switzerland) wherein the neuron-specific synapsin promoter has substituted the PGK

promoter. Generation of lentiviral particles was performed as previously described (*Yau et al., 2014*). Lentiviral transduction of cells was performed two hours after plating. The tetracycline-dependent expression was induced by adding 500 ng/ml doxycycline to the medium two days before imaging.

## Antibodies

Primary antibodies used for immunofluorescence in this study: mouse-IgG1 anti-Nestin (1:200; MAB5326, Millipore), rabbit anti-Ki67 (1:500; ab92742, Abcam), mouse-IgG2b anti-β3-Tubulin (1:400; T86605, Sigma), chicken anti-MAP2 (1:2000; ab5392, Abcam), rabbit anti-Trim46 (1:500; homemade [*van Beuningen et al., 2015*]), mouse-IgG1 anti-AnkG (1:200; 33–8800, Life Technologies), mouse-IgG1 anti-PanNav (1:200; S8809, Sigma), rat anti-Ctip2 (1:500, ab18465, Abcam), mouse-IgG2a anti-Sox2 (1:100, mab2018, R and D systems). Secondary antibodies used for immunofluorescence in this study: anti-rabbit Alexa 405 (A31556, Life Technologies), anti-mouse-IgG1 Alexa 488 (A21121, Life Technologies), anti-rat Alexa 488 (A11006, Life Technologies), anti-rabbit Alexa 568 (A11036, Life Technologies), anti-mouse Alexa 568 (A11031, Life Technologies), anti-mouse-IgG2b Alexa 594 (A21145, Life Technologies), anti-rabbit Alexa 647 (A21245, Life Technologies), anti-mouse Alexa 647 (A21236, Life Technologies), anti-mouse-IgG2a Alexa 647 (A21241, Life Technologies), anti-chicken Alexa 647 (A21449, Life Technologies). Primary antibodies used for western blot in this study: mouse-IgG1 anti-Actin (1:10.000; MAB1501R, Sigma), mouse-IgG1 anti-Nestin (1:5000; MAB535, Sigma), rabbit anti-Trim46 (1:5000; 377003, SySy), mouse-IgG2b anti-ß3-tubulin (1:5000; T8660, Sigma), rat anti-Ctip2 (1:5000; ab18465, Abcam), mouse-IgG2a anti-Sox2 (1:5000; mab2018, R and D systems), rabbit anti-GapDH (1:5000; G9545, Sigma), rabbit anti-Gap43 (1:5000; AB16053, Abcam). Secondary antibodies used for western blot in this study: goat-anti-mouse IRdye680LT (1:20.000; 926–68020, Li-Cor), goat-anti-rabbit IRdye800CW (1:15.000; 926–32211, Li-Cor), goat-anti-rat IRdye800CW (1:10.000; 926–32219, Li-Cor).

## Immunofluorescence

Cells were fixed for 5–10 min in PBS with 4% paraformaldehyde/4% sucrose at room temperature, and washed three times with PBS. For immunofluorescence stainings, fixed cells were sequentially incubated with primary and secondary antibodies dissolved in gelate dilution buffer (GDB; 0.2% BSA, 0.8 M NaCl, 0.5% Triton X-100, 30 mM phosphate buffer, pH 7.4), and washed three times with PBS after every antibody incubation step. Coverslips were mounted on glass slides using Vectashield mounting medium (Vector laboratories) with or without DAPI.

## Western blot

Cells were eluted at days 1, 5, and 14 in SDS/DTT sample buffer. For western blot analysis, samples were boiled at 95℃ for 10 min, loaded on 10% SDS-PAGE gels, and subsequently transferred to nitrocellulose membranes. The membranes were blocked using 2% BSA (bovine serum albumin, A7906, Sigma) in PBS/0.02% Tween-20 for at least 1 hr at room temperature. This blocking buffer was also used to dilute antibodies. Membranes were incubated with primary antibodies overnight at 4℃ and with secondary antibodies for 1 hr at room temperature. Membranes were washed three times with PBS/0.02% Tween-20 between incubation with primary and secondary antibodies, and again prior to scanning on Odyssey Infrared Imaging system (Li-Cor Biosciences).

## Microscopy

Fixed cells were imaged using a LSM700 confocal laser-scanning microscope (Zeiss) with a Plan-Apochromat 63x NA 1.4 oil DIC; EC Plan-Neofluar 40x, NA 1.3 Oil DIC; and a Plan-Apochromat 20x, NA 0.8 objective. Live-cell acquisition was performed using spinning-disk confocal microscopy on an inverted research microscope Nikon Eclipse Ti-E, equipped with a perfect focus system (Nikon) and a spinning disk-based confocal scanner unit (CSU-X1-A1, Yokogawa). The system was equipped with an ASI motorized stage with the piezo plate MS-2000-XYZ (ASI), Photometric Evolve Delta 512 EMCCD camera (Photometric) controlled by the MetaMorph 7.8 software (Molecular Devices), or Photometric PRIME BSI sCMOS camera (version USB 3) as upgrade of EMCCD and controlled by the MetaMorph 7.10 software (Molecular Devices). The system was also equipped with Plan Apo VC 60x NA 1.4 oil-immersion objective (Nikon) and S Fluor 100x NA 0.5–1.3 oil-immersion objective (Nikon)

for photoablation experiments. A 491 nm 100 mW Calypso (Cobolt) and a 561 nm 100 mW Jive (Cobolt) laser were used as the light sources. We used an ET-GFP filter set (49002, Chroma) for imaging of proteins tagged with GFP and an ET-mCherry filter set (49008, Chroma) for imaging of proteins tagged with tag-RFP. For the photoablation experiments we used an ILas system (Roper Scientific France/PICT IBiSA, Institut Curie, currently Gataca Systems) mounted on a Nikon Eclipse microscope described above. A 355 nm passively Q-switched pulsed laser (Teem Photonics) was used for the photoablation together with a S Fluor 100 $\times$ 0.5–1.3 NA oil objective (Nikon).

## Image quantification and analysis

### Quantifying neuronal differentiation and polarization

To measure neuronal differentiation and polarization over time, cells were identified using DAPI staining and scored to be positive or negative for the indicated NSC, neuron differentiation and axon markers.

### Quantification of stage 2a, 2b, 3a and 3b neurons

To determine the transition of neurodevelopmental stages over time, neurons were identified using DAPI and MAP2-positive immunofluorescence, and scored for neurodevelopmental stage 2a, 2b, 3a and 3b. Stage 2a and stage 2b neurons contained unipolar neurites of similar lengths. In stage 2a neurons, all neurites were negative for Trim46. In stage 2b neurons, one or more neurites were positive for Trim46. Stage 3a and stage 3b neurons were identified by the presence of a single elongated neurite, the future axon, that was at least twice as long as the other neurites. In stage 3a neurons, Trim46 appeared as distal non-continuous stretches at distal axons. In stage 3b neurons, Trim46 showed a dense accumulation at proximal axons.

## Live-cell imaging

For all live-cell imaging of microtubule dynamics without laser severing, time-lapse acquisition was performed using the 491 nm 100 mW Calypse (200 ms exposure) and 561 nm 100 mW Jive (200 ms exposure) with one frame per second (fps) for 5 min. 16-bit images were projected onto the EMCCD chip with intermediate lens 2.0X (Edmund Optics) at a magnification of 0.111 µm/pixel at 60x, or onto the sCMOS chip with no intermediate lens at a magnification of 0.150 µm/pixel at 60x. For all live-cell imaging of microtubule dynamics with laser severing, time-lapse acquisition was performed using the 491 nm 100 mW Calypse (50–200 ms exposure, depending on the expression level) and 561 nm 100 mW Jive (50–200 ms exposure, depending on the expression level) with 1 fps for 3 min, and photoablation was induced after 30 s. 16-bit images were projected onto the sCMOS chip with no intermediate lens at a magnification of 0.063 µm/pixel at 100x. All imaging was performed in full conditioned differentiation (day 5) or maintenance (day 7 or 13) medium for hiPSC-derived neuron cultures, and cells were kept at 37°C with 5% $CO_2$ using a stage top incubator (model INUBG2E-ZILCS, Tokai Hit). For analysis, kymographs were generated using the FIJI plugin KymoResliceWide v.0.4 (https://github.com/ekatrukha/KymoResliceWide; *Katrukha, 2017*), and parameters of microtubule plus-end dynamics were determined by manually tracing microtubule growth events.

## Electrophysiology

A 12 mm coverslip containing hiPSC-derived neurons (7–14 days after plating) was transferred to the microscope recording chamber before the start of each experiment. Coverslips were continuously perfused with carbonated (95% $O_2$, 5% $CO_2$) artificial cerebrospinal fluid (ACSF, in mM: 126 NaCl, 3 KCl, 2.5 $CaCl_2$, 1.3 $MgCl_2$, 26 $NaHCO_3$, 1.25 $NaH_2PO_4$, 20 glucose; with an osmolarity of ~310 mOsm/L) at a rate of approximately 1 ml/min. As an acute change in extracellular osmolarity has previously been reported to affect the excitable properties of neurons, an extra medium refreshment was done the day before recording (*Pasantes-Morales, 1996*). Bath temperature was monitored and maintained at 30–32°C throughout the experiment. Recording pipettes (resistance of 4–7 MΩ) were pulled from thick-walled borosilicate glass capillaries (World Precision Instruments) and filled with internal solution (in mM: 140 K-gluconate, 4 KCl, 0.5 EGTA, 10 HEPES, 4 MgATP, 0.4 NaGTP, 4 $Na_2$-Phosphocreatine; with pH 7.3 and osmolarity 295 mOsm/L), containing 30 µM Alexa 568 (Thermo Fisher Scientific) to facilitate visualization of cells. For post hoc cellular identification, biocytin was included in the internal solution. On an upright microscope, hiPSC-derived neurons were

visually identified with a 60x water immersion objective (Nikon NIR Apochromat; NA 1.0) and selected for whole-cell somatic patch clamp recordings. Cells were kept at a baseline holding potential of −60 or −70 mV in both voltage and current clamp throughout the recording. Recordings were acquired with an Axopatch 200B amplifier (Molecular Devices) using pClamp 10 software. Data were analyzed with Clampfit 10.7 software and custom-written MATLAB scripts.

## Sample preparation RNA sequencing

~100,000 hiPSC-derived NPCs were plated per well for bulk RNA sequencing samples. Prior to sample preparation, all equipment and surfaces were cleaned with RNaseZap (Sigma-Aldrich). Biological replicates (samples derived from NSCs that have been independently expanded and differentiated, but prepared and analyzed under the same conditions) and technical replicates (separate samples harvested in parallel from the same biological replicate) of hiPSC-derived neurons were harvested at three different timepoints of differentiation (days 1, 3, and 7) with 200 µl Trizol (Invitrogen) per sample and stored at −80℃ until sequencing. RNA extraction, cDNA library preparation (CEL-Seq2 protocol), quality control for aRNA and cDNA, and sequencing on a NextSeq500 High output 1 × 75 bp paired end run with 2% sequencing depth were performed by Single Cell Discoveries (Utrecht, The Netherlands).

## Bioinformatic analysis RNA sequencing

Mapping to reference transcriptome Hg19 was performed by Single Cell Discoveries (Utrecht, The Netherlands). The following investigations were done in R statistical software (*R Development Core Team, 2020*) with the use of packages ggplot2 (*Wickham and Sievert, 2016*) and pheatmap (*Kolde, 2019*; *Figure 2—source code 1*). The raw read counts were normalized to reads per million for each gene. Genes observed in less than five samples were excluded from further analysis. To determine differential expression a linear regression ANOVA model was used where gene expression is explained by timepoint + biological replicate and technical replicate (nested in the biological replicate). To obtain the differences (in both p-value and effect between the timepoints) a Tukey-test was performed for each gene using the same model. We corrected for multiple-testing by pairwise comparison using the p.adjust() function in R with the 'BH' setting (*R Development Core Team, 2020*). Genes with an adjusted p-value<0.05 were used for further investigation. GO enrichment was done using the hypergeometric test phyper() in R (*R Development Core Team, 2020*), the set of GO terms was obtained from Ensembl Biomart for Human genes version GRCh38.p13. The RNA sequencing data have been deposited in NCBI's Gene Expression Omnibus and are accessible through GEO Series accession number GSE155212 (https://www.ncbi.nlm.nih.gov/geo/query/acc.cgi?acc=GSE155212) (*Edgar et al., 2002*; *Barrett et al., 2013*). Comparison between transcriptomics and proteomics was done by selecting genes present in both datasets based on their public name.

## Sample preparation for mass spectrometry (TMT labeling)

Replicates of hiPSC-derived neurons were harvested with lysis buffer (8 M Urea, 50 mM ammonium bicarbonate (Sigma), EDTA-free protease inhibitor Cocktail (Roche)) at three distinct differential time points (days 1, 3, and 7). Lysates were sonicated on ice with a Bioruptor (Diagenode) and cleared by centrifugation at 2500 *g* for 10 min. The protein concentration of the samples was determined by Bradford assay. Per sample 100 µg of proteins were reduced (5 mM DTT, 55 ℃, 1 hr), alkylated (10 mM Iodoacetamide, 30 min in the dark) and sequentially digested by LysC (Protein-enzyme ratio 1:50, 37 ℃, 4 hr) and trypsin (Protein-enzyme ratio 1:50, 37 ℃, overnight). After digestion (overnight), formic acid (final concentration 3%) was used to acidify the samples and resulting peptides were afterwards desalted with Sep-Pak C18 columns (Waters). Samples were labeled with stable isotope TMT-6plex labeling, according to manufacturer's instruction (Thermo Fisher Scientific). In short, peptides were resuspended in 80 µl of 50 mM HEPES buffer, 12.5% ACN (pH 8.5), while TMT reagents were dissolved in 50 µl anhydrous ACN. We added 25 µl of each dissolved TMT reagent to a correspondent sample according to the following scheme:

> day 1 (replicate A)=TMT-126/day 1 (replicate B)=TMT-129
> day 3 (replicate A)=TMT-127/day 3 (replicate B)=TMT-130
> day 7 (replicate A)=TMT-128/day 7 (replicate B)=TMT-131

Following incubation (room temperature) for 1 hr, the reaction was quenched with 5% hydroxyl-amine. Differentially TMT-labeled peptides were mixed in equal ratios and dried in a vacuum concentrator.

## Peptide fractionation

Mixed TMT-labeled peptides were solubilized in 10 mM ammonium hydroxide, pH 10.0 and subsequently fractionated using basic pH reverse phase HPLC. Peptides were loaded on a Gemini 3 μm C18 110A 100 × 1.0 mm column (Phenomenex) using an Agilent 1100 pump equipped with a degasser and a photodiode array (PDA) detector. Peptides were concentrated on the column at 100 μl/min using 100% buffer A (10 mM ammonium hydroxide, pH 10) after which the fractionation gradient was applied as follow: 5% solvent B (10 mM ammonium hydroxide in 90% ACN, pH 10) to 30% B in 53 mins, 70% B in 7 min and increased to 100% B in 3 min at a flow rate of 100 μl/min. In total 60 fractions of 1 min were collected using an Agilent 1260 infinity fraction collector and then concatenated into 12 final fractions. Collected fractions were vacuum-dried, reconstituted in 5% formic acid/5% DMSO and stored at −80°C prior to mass spectrometry analysis.

## Mass spectrometry analysis

We analyzed the samples on an Orbitrap Q-Exactive HF mass spectrometer (Thermo Fisher Scientific) coupled online to an Agilent UPLC 1290 system (Agilent Technologies). Peptides were loaded onto a trap column (Reprosil C18, 3 μm, 2 cm ×100 μm; Dr. Maisch) and separated on an analytical column (Poroshell 120 EC-C18, 2.7 μm, 50 cm x 75 μm; Agilent Technologies). Peptides were trapped for 10 min at 5 μl/min in solvent A (0.1M acetic acid in H20) and then separated at a flow rate of approximately 300 nl/min (split flow from 0.2 ml/min) by applying a 120 min linear gradient as follows: 13% up to 40% solvent B (0.1M acetic acid in 80% ACN) in 95 min, 40–100% in 3 min and finally 100% for 1 min. The mass spectrometer was operated in data-dependent acquisition mode. Full MS spectra from m/z 375–1600 were acquired at a resolution of 60.000 with an automatic gain control (AGC) target value of 3e6 and maximum injection time (IT) of 20 ms. The 15 most intense precursor ions were selected for HCD fragmentation. HCD fragmentation was performed at a normalized collision energy (NCE) of 32%. MS/MS spectra were obtained at a 30.000 resolution with an AGC target of 1e5 and maximum injection time (IT) of 50 ms. Isolation window was set at 1.0 m/z and dynamic exclusion to 16.0 s.

## Data processing proteomics

Raw MS files were processed for data analysis with Proteome Discoverer 1.4 (Thermo Fisher Scientific). A database search was performed using the Swissprot *Homo sapiens* database and Mascot (version 2.5.1, Matrix Science, UK) as search engine. Carbamidomethylation of cysteines was set as a fixed modification, and oxidation of methionine, acetylation at the N-termini, TMT-6plex of lysine residues and TMT-6plex at the peptide N-termini were set as variable modifications. Trypsin was set as cleavage specificity, allowing a maximum of two missed cleavages. Data filtering was performed using percolator, resulting in 1% false discovery rate (FDR). Additional filters were search engine rank one and mascot ion score >20. Only unique peptides were included for quantification and the obtained TMT ratios were normalized to the median. Common contaminant proteins (such as keratins and albumin) were removed from the list. The mass spectrometry proteomics data have been deposited to the ProteomeXchange Consortium via the PRIDE partner repository with the dataset identifier PXD020227 (*Perez-Riverol et al., 2019*; *Deutsch et al., 2020*).

## Bioinformatic analysis proteomics

All mass spectrometry data were analyzed using R statistical software (*R Development Core Team, 2020*; *Figure 2—source code 1*). To infer protein dynamics upon differentiation, TMT reporter intensity values of hiPSC neurons at time point day 3 and day 7 were normalized to their correspondent day one or alternatively day 7 was normalized to day 3. TMT generated ratios (previously normalized to the median) were then log2-transformed. A log2-transformed mean of the TMT-ratios of the individual replicates was calculated. Proteins with less than three peptides used for TMT quantification or with a reporter ion variability >100% in at least one TMT-ratio (high reporter ions

variability) or with a median log2 fold-change >0.4 between the replicates in at least one TMT-ratio (high replicate variability) were excluded from the analyses. Good correlation of replicates was assessed by comparing TMT ratios of all quantified proteins at different time points using Pearson correlation. Proteins with an absolute log2 fold-change >0.3 between day 3 and day 1, between day 7 and day 3, or between day 7 and day 1 were considered significantly regulated. Only significantly regulated proteins were subjected to cluster analysis by using K-means clustering in R. Functional enrichment analysis within different clusters of expression profiles was performed using gProfilerR package in R (*Raudvere et al., 2019* #173}. Network analyses were performed using the GeneMania plugin {*Montojo et al., 2010*) within Cytoscape (*Montojo et al., 2010*). Heatmaps in the figures were prepared applying hierarchical clustering using Euclidean distance.

## Statistical analysis

Data processing and statistical analyses were performed using Prism GraphPad (version 8.0) software. All statistical details of performed experiments, such as performed statistical tests, and values of N and n, are described in the figure legends. The number of repeated independent experiments is defined as N, and the number of individual data points as n. Significance was defined as: ns = not significant or $p > 0.05$, * for $p < 0.05$, ** for $p < 0.005$, *** for $p < 0.001$. Graphs represent mean ± SEM.

## Acknowledgements

We thank Dr. Didier Trono for the lentiviral vector, and Nicky Scheefhals for her constructive feedback on the manuscript. This work was supported by the Netherlands Organization for Scientific Research (NWO-ALW-VICI, 865.10.010, CCH), the Netherlands Organization for Health Research and Development (ZonMW-TOP, 912.16.058, CCH), the European Research Council (ERC) (ERC-consolidator, 617050, CCH), and the research program of the Foundation for Fundamental Research on Matter (FOM, #16NEPH05, CJW).

## Additional information

### Competing interests

Casper C Hoogenraad: Employee of Genentech, Inc, a member of the Roche group. The other authors declare that no competing interests exist.

### Funding

| Funder | Grant reference number | Author |
| --- | --- | --- |
| European Research Council | 617050 | Feline W Lindhout Riccardo Stucchi |
| ZonMw | 912.16.058 | Robbelien Kooistra |
| Foundation for Fundamental Research on Matter | 16NEPH05 | Lotte J Herstel |

The funders had no role in study design, data collection and interpretation, or the decision to submit the work for publication.

### Author contributions

Feline W Lindhout, Conceptualization, Data curation, Formal analysis, Investigation, Writing - original draft; Robbelien Kooistra, Sybren Portegies, Conceptualization, Data curation, Formal analysis, Investigation, Visualization, Writing - original draft; Lotte J Herstel, Data curation, Formal analysis, Investigation, Visualization, Writing - original draft; Riccardo Stucchi, Formal analysis, Validation, Visualization; Basten L Snoek, Formal analysis, Investigation, Writing - review and editing; AF Maarten Altelaar, Resources, Software, Supervision, Writing - review and editing; Harold D MacGillavry, Supervision, Writing - review and editing; Corette J Wierenga, Funding acquisition, Visualization, Writing - review and editing; Casper C Hoogenraad, Funding acquisition, Writing - review and editing

## Author ORCIDs

Feline W Lindhout [ID] https://orcid.org/0000-0001-5075-5434
Robbelien Kooistra [ID] https://orcid.org/0000-0003-4576-8964
Sybren Portegies [ID] https://orcid.org/0000-0003-1654-4574
Lotte J Herstel [ID] http://orcid.org/0000-0002-2839-4641
Casper C Hoogenraad [ID] https://orcid.org/0000-0002-2666-0758

## Decision letter and Author response

Decision letter https://doi.org/10.7554/eLife.58124.sa1
Author response https://doi.org/10.7554/eLife.58124.sa2

# Additional files

## Supplementary files

• Transparent reporting form

## Data availability

Raw RNA sequencing data and proteomics data are deposited in online repositories. RNA sequencing data: https://www.ncbi.nlm.nih.gov/geo/query/acc.cgi?acc=GSE155212. Proteomics data: https://www.ebi.ac.uk/pride/archive/projects/PXD020227. All processed bioinformatics data are included in Figure 2—source data 1–6.

The following datasets were generated:

| Author(s) | Year | Dataset title | Dataset URL | Database and Identifier |
|---|---|---|---|---|
| Lindhout FW, Kooistra R, Portegies S, Herstel LJ, Stucchi R, Snoek BL, Altelaar M, MacGillavry HD, Wierenga CJ, Hoogenraad CC | 2020 | Quantitative mapping of transcriptome dynamics during polarization of human iPSC-derived neurons | https://www.ncbi.nlm.nih.gov/geo/query/acc.cgi?acc=GSE155212 | NCBI Gene Expression Omnibus, GSE155212 |
| Lindhout FW, Kooistra R, Portegies S, Herstel LJ, Stucchi R, Snoek BL, Altelaar M, MacGillavry HD, Wierenga CJ, Hoogenraad CC | 2020 | Quantitative mapping of transcriptome dynamics during polarization of human iPSC-derived neurons | https://www.ebi.ac.uk/pride/archive/projects/PXD020227 | PRIDE, PXD020227 |

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
