## [Decision Letter]

**Acceptance summary:**

This study reports neuronal polarization in human iPSC derived neurons, by the characterization of electrophysiological responses, and systematically profiles transcriptomic and proteomic dynamics during early neuronal developmental stages. The results indicate extensive remodeling of the neuron transcriptome and proteome during development. They also identify a distinct stage in axon development marked by an increase in microtubule remodeling and apparent relocation of the axon initial segment from the distal to the proximal axon.

**Decision letter after peer review:**

Thank you for submitting your article "Quantitative mapping of transcriptome and proteome dynamics during polarization of human iPSC-derived neurons" for consideration by *eLife*. Your article has been reviewed by three peer reviewers, one of whom is a member of our Board of Reviewing Editors, and the evaluation has been overseen by Marianne Bronner as the Senior Editor. The reviewers have opted to remain anonymous.

The reviewers have discussed the reviews with one another and the Reviewing Editor has drafted this decision to help you prepare a revised submission.

Summary:

In this paper Lindhout et al. studied neuronal polarization in human iPSC derived neurons, by the characterization of electrophysiological responses, and systematically profiled transcriptomic and proteomic dynamics during early neuronal developmental stages. As expected they found extensive remodeling of the neuron transcriptome and proteome during development. They also described a distinct stage in axon development marked by an increase in microtubule remodeling and apparent relocation of the axon initial segment from the distal to the proximal axon.

This is a comprehensive characterization and quantitative map of transcriptome and proteome dynamics of early-stage development of human iPSC derived neurons. However, the overall data remain in a descriptive stage and the findings describe the expected overall transcriptome and proteome changing during neuronal development already described in other models. In addition, the lack of some critical experiments does not allow us to consider the data presented fully complete for a compelling study.

Essential revisions:

1) All the experiments have been done using only one iPSC line, but considering the high heterogeneity of the different human iPSC lines, the obtained results should be confirmed, at least for some data, with other iPSC lines. Indeed the authors claim that the observed intermediate step of distal AIS protein accumulation is unique to human neurons. These data can be considered valid only if confirmed in other independent iPSC lines.

2) Another important aspect is that the authors used only one method for neuronal differentiation. In the literature, however, many protocols have been described. Thus the obtained results might not be possible to be extended if NPCs are differentiated with other methodologies. This aspect reduces considerably the impact of the paper.

---

## [Author Response]

Essential revisions:1) All the experiments have been done using only one iPSC line, but considering the high heterogeneity of the different human iPSC lines, the obtained results should be confirmed, at least for some data, with other iPSC lines. Indeed the authors claim that the observed intermediate step of distal AIS protein accumulation is unique to human neurons. These data can be considered valid only if confirmed in other independent iPSC lines.

We agree with the reviewers that it is important to confirm some data, and in particular the identified intermediate axon developmental step, in other iPSC lines. To address this, we carefully selected more iPSC lines to get a total of three independent iPSC cell lines of donors with mixed gender, age (newborn and 74 years) and using different starting material (cord blood CD34+ cells and fibroblasts). We observed successful AIS assembly as well as action potential (AP) firing upon current stimulation in all cell lines, indicating that iPSCs derived from different donors all developed into polarized and functional neurons (Figure 1—figure supplement 1E, F, H, I). Importantly, the newly identified intermediate axon developmental step characterized by distal AIS protein accumulation was also consistently observed in all cell lines, thereby confirming our initial findings (Figure 3—figure supplement 1A, B).

2) Another important aspect is that the authors used only one method for neuronal differentiation. In the literature, however, many protocols have been described. Thus the obtained results might not be possible to be extended if NPCs are differentiated with other methodologies. This aspect reduces considerably the impact of the paper.

We thank the reviewers for raising this point. Indeed, it is relevant to validate some of our key findings using other neuronal differentiation methodologies. To address this, we used two additional protocols that are based on either induced neuronal differentiation – a largely adjusted version of our standard protocol – or spontaneous neuronal differentiation (please see the Materials and methods section for details). As with our standard protocols, differentiated neurons showed successful AIS assembly as well as action potential firing upon current stimulation with both other differentiation protocols (Figure 1—figure supplement 1E, F, H, I). Moreover, we steadily observed that developing neurons proceeded through axon developmental stage 3a, marked by the appearance of AIS proteins in distal axons, using any of the neuronal differentiation protocols (Figure 3—figure supplement 1A, B). Thus, we report similar results in NSCs differentiated using three different methodologies, thereby largely increasing the possibility to extend our findings.